# Small Nucleolar RNAs Determine Resistance to Doxorubicin in Human Osteosarcoma

**DOI:** 10.3390/ijms21124500

**Published:** 2020-06-24

**Authors:** Martina Godel, Deborah Morena, Preeta Ananthanarayanan, Ilaria Buondonno, Giulio Ferrero, Claudia M. Hattinger, Federica Di Nicolantonio, Massimo Serra, Riccardo Taulli, Francesca Cordero, Chiara Riganti, Joanna Kopecka

**Affiliations:** 1Department of Oncology, University of Torino, 1026 Torino, Italy; martina.godel@edu.unito.it (M.G.); deborah.morena@unito.it (D.M.); preeta.ananthanarayanan@unito.it (P.A.); ilaria.buondonno@unito.it (I.B.); federica.dinicolantonio@unito.it (F.D.N.); riccardo.taulli@unito.it (R.T.); 2Department of Computer Science, University of Torino, 10149 Torino, Italy; giulio.ferrero@unito.it (G.F.); francesca.cordero@unito.it (F.C.); 3Department of Clinical and Biological Sciences, University of Torino, 10043 Orbassano, Italy; 4Laboratory of Experimental Oncology, Pharmacogenomics and Pharmacogenetics Research Unit, IRCCS Istituto Ortopedico Rizzoli, 40136 Bologna, Italy; claudia.hattinger@ior.it (C.M.H.); massimo.serra@ior.it (M.S.); 5Candiolo Cancer Institute, FPO–IRCCS, 10060 Candiolo, Italy

**Keywords:** osteosarcoma, doxorubicin, chemoresistance, small nucleolar RNAs

## Abstract

Doxorubicin (Dox) is one of the most important first-line drugs used in osteosarcoma therapy. Multiple and not fully clarified mechanisms, however, determine resistance to Dox. With the aim of identifying new markers associated with Dox-resistance, we found a global up-regulation of small nucleolar RNAs (snoRNAs) in human Dox-resistant osteosarcoma cells. We investigated if and how snoRNAs are linked to resistance. After RT-PCR validation of snoRNAs up-regulated in osteosarcoma cells with different degrees of resistance to Dox, we overexpressed them in Dox-sensitive cells. We then evaluated Dox cytotoxicity and changes in genes relevant for osteosarcoma pathogenesis by PCR arrays. SNORD3A, SNORA13 and SNORA28 reduced Dox-cytotoxicity when over-expressed in Dox-sensitive cells. In these cells, GADD45A and MYC were up-regulated, TOP2A was down-regulated. The same profile was detected in cells with acquired resistance to Dox. GADD45A/MYC-silencing and TOP2A-over-expression counteracted the resistance to Dox induced by snoRNAs. We reported for the first time that snoRNAs induce resistance to Dox in human osteosarcoma, by modulating the expression of genes involved in DNA damaging sensing, DNA repair, ribosome biogenesis, and proliferation. Targeting snoRNAs or down-stream genes may open new treatment perspectives in chemoresistant osteosarcomas.

## 1. Introduction

The treatment of osteosarcoma, one the most frequent tumors in childhood, relies on conservative surgery and neo-adjuvant or adjuvant chemotherapy, mostly based on cisplatin, methotrexate, and doxorubicin (Dox). In recent years, several targeted-therapies or immunotherapies have been evaluated in clinical trials, but these treatments did not significantly improve patients’ outcome [1,2,3]. Chemotherapy still remains the first treatment option, although the constitutive or acquired resistance is a strong limitation [3].

Chemoresistance is mediated by decreased drug influx or increased efflux, enhanced drug inactivation, improved efficacy of deoxyribonucleic acid (DNA) repair machinery, prevalence of pro-survival pathways over pro-apoptotic ones, presence of highly resistant cancer stem cells [4]. The main mechanism of resistance to Dox in osteosarcoma is the presence of the ATP binding cassette transporter B1 (ABCB1), also known as P-glycoprotein (Pgp), which effluxes Dox outside the cells limiting its intracellular accumulation and toxicity [5]. Pgp has been clinically validated as a robust biomarker predictive of patients’ outcome and response to first-line chemotherapy [6,7,8], but resistance to Dox is a multifactorial process in osteosarcoma [4].

Recently, long non-coding RNAs (lncRNAs) and microRNAs (miRNAs) have been pointed out as key determinants in osteosarcoma tumorigenesis, progression, invasion and drug resistance [9,10]. For instance, the lncRNA lung cancer associated transcript 1 (LUCAT1) sensitizes osteosarcoma cells to methotrexate by sponging miR-200c, a Pgp inducer [11]. Similarly, by sponging miR-645, the lncRNA long intergenic non-protein coding RNA 161 (LINC00161) down-regulates the pro-apoptotic factor interferon-induced protein with tetratricopeptide repeats 2 (IFIT-2), increasing the resistance to cisplatin-induced apoptosis [12]. Members of the lncRNA small nucleolar RNA host gene (SNHG) family have been implicated in the chemoresistance of osteosarcoma, as well. SNHG12 is higher in Dox-resistant osteosarcoma cells than in sensitive cells and correlates with lower patients’ survival: specifically, SNHG12 induces resistance by up-regulating the mir320a/myeloid cell leukemia 1 (MCL1) axis that protects from the apoptosis induced by Dox [13]. SNHG16 favors osteosarcoma progression, invasion and resistance to cisplatin by sponging miR-16 and down-regulating the autophagy-related 4B (ATG4B) protein, impairing both cisplatin-induced autophagy and apoptosis [14].

If lncRNAs have been extensively studied in the progression and resistance of osteosarcoma, less attention has been payed to small nucleolar RNAs (snoRNAs). Structurally, snoRNAs are divided into two families: (i) the box H/ACA snoRNAs (SNORA), characterized by hairpins complementary to ribosomal RNA (rRNA), separated by the H-box region and terminated with an ACA motif; (ii) the box C/D snoRNAs (SNORD) that have a terminal stem and sequence motifs (termed C, C’, D’ and D) complementary to the target rRNA [15]. SnoRNAs have been identified as organizers of nucleolar RNA [16] or as guides for methylation, pseudo-uridylation and acetylation of rRNA [15]. Now, it is clear that snoRNAs are also exported from the nucleolus [17], mediate mRNA processing, compete with other ncRNAs or proteins for binding functional sites on mRNA [15,18]. In addition, they modulate exome recruitment and chromatin remodelling [15,18]. Some snoRNAs are included at 5′-end of lncRNAs [16]; others–belonging to SNORD subfamily-produce regulatory RNAs, termed sno-miRNAS [19], enlarging the number of biological processes controlled. Most snoRNAs are contained within introns of pre-mRNA [17]: they are transcribed by RNA polymerase II and require the activity of specific endo/exonucleases to mature [20]. The set of splicing and maturation enzymes is different in each organ [21,22], as well as in tumors [23], implying a highly variable pattern of snoRNAs in different tissues. In cancers, snoRNAs have either an oncogene or an onco-suppressor function: they have been implicated in cell proliferation, apoptosis and metastasis, as a consequence of the up- or down-regulation of the respective host genes [24,25]. To the best of our knowledge, linkages between snoRNAs and drug resistance have not been reported yet.

Dox is a strong inducer of nucleolar stress because the DNA damage elicited by the drug disaggregates the nucleolar structure, impairs the ribosome biogenesis, promotes the translocation of the cytoprotective protein nucleophosmin from nucleolus to cytoplasm [26]. Dox exposure also activates a broad transcriptional program that decreases aminoacid biosynthesis, tRNA aminoacylation and general translation [27]. In human osteosarcoma U-2OS cells, Dox-induced DNA damage increases the ubiquitination of ribosomal proteins of both 40 S and 60 S subunits [28]. Moreover, Dox inhibits mammalian target of rapamycin (mTOR) activity: as a consequence, the phosphorylation of p70 S6 kinase (p70 S6K) and of the eukaryotic initiation factor 4E-binding protein 1 (eIF4E-BP1) are reduced. These events impair the p70 S6K-mediated phosphorylation of ribosomal proteins and sequester the cap-binding protein eIF4E, respectively [29]. Another ribosome-related mechanism of Dox cytotoxicity is the cytoplasmatic export of C/D snoRNAs contained in the introns of the ribosomal protein L13a (Rpl13a), a physiological contributor of cell death in response to oxidative stress [30]. The sum of the decreased ribosome biogenesis, increased ribosomal protein degradations, and lower efficiency of the initiation complex dramatically reduce protein translation and trigger apoptosis in Dox-sensitive cells. On the other hand, resistance to Dox has been associated to the increased expression of specific ribosome components. In U-2OS cells, Dox-induced DNA damage and cell cycle arrest decrease the expression of the nucleolar protein ribosomal RNA processing 12 homolog (RRP12) that is involved in the export and maturation of 40 S and 60 S ribosome components. By contrast, RRP12 overexpression determines Dox resistance [31]. Similarly, the aberrant expression of the receptor for activated C-kinase 1 (RACK1) and its ribosomal localization induces resistance to Dox, by phosphorylating eIF4E that triggers the preferential translation of pro-survival factors [32].

Considering the pleiotropic biological processes controlled by lncRNAs in osteosarcoma, the physiological role of snoRNAs as modulators of ribosomal functions and the inhibitory effects of Dox on ribosome biogenesis and protein translation, we explored the panorama of snoRNAs in Dox-sensitive and Dox-resistant human osteosarcoma cells, with the goal of identifying possible cause-effect relations between the expression of specific snoRNAs and the resistance to Dox.

## 2. Results

### 2.1. snoRNA Family is Up-Regulated in Doxorubicin-Resistant Osteosarcoma Cells

Transcriptome profiling of Dox-sensitive U-2OS cells and of their resistant variants (U-2OS/DX30, U-2OS/DX100 and U-2OS/DX580), characterized by progressively increasing resistance to Dox [26], was performed. Appendix A shows the genes significantly modulated in resistant variant. As expected, specific genes involved in drug resistance, such as *ABCB1*, encoding for Pgp, were found up-regulated in Dox-resistant cells. Interestingly, several snoRNAs were included within the significantly modulated genes.

The analysis of the 277 snoRNAs probes present in the array (Appendix A) confirmed that 160 snoRNAs were up-regulated by Dox treatment in U-2OS cells or at least in one resistant variant (Figure 1A). Specifically, 127, 132, 23, and 1 snoRNAs were significantly regulated (*p* < 0.05) by Dox-treatment in U-2OS, U-2OS/DX30, U-2OS/DX100 and U-2OS/DX580 cells, respectively (Figure 1B). Furthermore, considering the comparison of the expression levels of these snoRNAs between resistant and sensitive cells, 139 out of the 160 Dox-regulated snoRNAs were up-regulated at least in one resistant variant (Figure 1C).

### 2.2. SNORD3A, SNORA13 and SNORA28 Are Ontologically and Functionally Related to Doxorubicin Resistance in Osteosarcoma Cells

We re-validated the expression of each snoRNAand host genes of known biological meaning (https://www.genecards.org/) by RT-PCR in Dox-sensitive osteosarcoma cells and in the resistant variants. In 3 cases–*SNORD3A* and organic solute carrier partner 1 (*OSCP1*), *SNORA13* and erythrocyte membrane protein band 4.1 like 4A (*EPB41L4A*), *SNORA28* and eukaryotic translation initiation factor-5 (*EIF5*)–both snoRNAs and host genes were progressively up-regulated with the increase of Dox-resistance (Figure 2A), with a strong linear correlation between the expression level of snoRNA and host gene mRNAs (Figure 2B). The levels of OSCP1, EPB41L4A and EIF5 proteins varied accordingly to the respective mRNAs (Figure 2C). This trend was not cell line-specific: indeed *SNORD3A*, *SNORA13* and *SNORA28* and their respective host genes were progressively up-regulated also in the Dox-resistant variants Saos-2/DX30, Saos-2/DX100 and Saos-2/DX580 (Appendix A) derived from Dox-sensitive Saos-2 cells, a second human osteosarcoma cell line [33]. These robust correlations led us to hypothesize that these 3 snoRNAs were processed and functioning in Dox-resistant variants. We thus investigated if they may play a role in determining the resistance to Dox of osteosarcoma cells.

The Gene Ontology Analysis, available for *SNORD3A*, identified RNA processing, protein synthesis initiation and chromosome segregation as the biological processes more strictly controlled (Appendix A), in line with the already known physiological role of this snoRNA [34]. Interestingly, the “doxorubicin DB00997 human GSE58074 sample 3180” was the “Drug perturbation” dataset most correlated with *SNORD3A* (Appendix A). No association data were available for *SNORA13* and *SNORA28*.

To functionally validate the role of *SNORD3A*, *SNORA13* and *SNORA28* as inducers of Dox resistance, we over-expressed them in Dox-sensitive U-2OS cells (Figure 3A). The expression levels of ectopic snoRNAs in U-2OS cells was in the range of U-2OS/DX30 and U-2OS/DX100 variants (Figure 2A and Figure 3A). Dox intracellular accumulation in snoRNA-over-expressing cells remained similar to parental U-2OS cells (Figure 3B). Also, the amount of Pgp in *SNORD3A*-, *SNORA13*- and *SNORA28*-expressing cells was comparable to U-2OS cells, and lower than the amount of Pgp detected in resistant variants (Figure 3C). However, snoRNA-over-expressing cells were more resistant to the cytotoxic effects of Dox. As expected, Dox significantly increased the release of lactate dehydrogenase (LDH) (Figure 3D), a marker of cell necrosis, and decreased cell viability (Figure 3E,F) in U-2OS cells, not in U-2OS/DX30 and U-2OS/DX100 variants. Interestingly, *SNORD3A*-, *SNORA13*-and *SNORA28*-over-expressing cells did not show any increase in LDH release (Figure 3D) or decrease in cell viability (Figure 3E,F), suggesting that each snoRNA can induce resistance to Dox in sensitive cells.

### 2.3. SNORD3A, SNORA13 and SNORA28 Contribute to Doxorubicin Resistance by Up-Regulating GADD45A and c-MYC, and Down-Regulating Topoisomerase 2A

We next compared the expression of genes relevant for osteosarcoma pathogenesis and progression [1] (Appendix A) in U-2OS cells, either wild-type or over-expressing *SNORD3A*, *SNORA13* and *SNORA28*, and in U-2OS/DX30 and U-2OS/DX100 variants. As shown in Figure 4A, Dox-resistant variants and snoRNA-overexpressing Dox-sensitive cells shared the up-regulation of growth arrest and DNA-damage-inducible α (*GADD45A*), a sensor of stressing conditions that increases after DNA damage [35], and *c-MYC*, an oncogene commonly amplified or mutated in cancer that favors cell cycle progression [36]. By contrast, the cells had down-regulated topoisomerase 2A (*TOP2A*), an enzyme catalyzing the temporarily breaking of DNA double strands followed by their rejoining during DNA replication or transcriptions [37]. The gene modulation was confirmed by the changes in the protein expression (Figure 4B,C).

To evaluate if the changes in GADD45A, MYC, and TOP2A levels were responsible for the resistance to Dox, induced by the three snoRNAs identified, we silenced *GADD45A* and *MYC*, and overexpressed *TOP2A* in U-2OS cells overexpressing *SNORD3A*, *SNORA13*, and *SNORA28* (Figure 5A). These changes did not alter the intracellular content of Dox that remained similar to U-2OS/DX30 and U-2OS/DX100 variants (Appendix A). Of note, the silencing of *GADD45A* and *MYC*, and the over-expression of *TOP2A* re-induced the toxicity of Dox in U-2OS cells expressing *SNORD3A*, *SNORA13* and *SNORA28*, as indicated by the increase of LDH release (Figure 5B) and decrease in cell viability (Figure 5C; Appendix A). Also, in U-2OS/DX30 and U-2OS/DX100 cells, the same genetic manipulations restored the cytotoxic effect of Dox (Figure 5C; Appendix A). These data suggest that: (i) *SNORD3A*, *SNORA13* and *SNORA28* induce resistance to Dox by up-regulating *GADD45A* and *MYC*, and/or by down-regulating *TOP2A*; (ii) changes in *GADD45A*, *MYC* and *TOP2A* determine the acquired resistance to Dox, observed in U-2OS/DX30 and U-2OS/DX100 variants.

## 3. Discussion

Until now, the expression level of Pgp is the most robust and clinically recognized factor predictive of response to Dox in osteosarcoma [6,8,38]. Chemoresistance, however, relies on pleiotropic factors in this tumor [4]. Identifying new biomarkers of Dox-resistance is a challenge still open. We addressed this issue by analyzing the whole-genome expression profile of human Dox-sensitive U-2OS cells and U-2OS/DX30, U-2OS/DX100 and U-2OS/DX580 variants, characterized by increasing resistance to Dox, acquired after a stepwise selection in Dox-containing medium [26]. This process that mimics the development of resistance acquired in patients exposed to cumulative doses of Dox, generated two variants–U-2OS/DX30 and U-2OS/DX100–with moderate resistance to the drug [33,39] and levels of Pgp comparable to those observed in the majority of clinical specimens [6,7,8]. U-2OS/DX580 cells exacerbated the expression of Pgp because of the high concentration of Dox stably present in their culture medium and were representative of rare and strongly resistant osteosarcomas.

In our unsupervised analysis, we found a general up-regulation of the snoRNA family as a typical feature of Dox-resistant variants. Interestingly, the up-regulation of single snoRNAs, as well as the number of snoRNAs up-regulated, was higher in U-2OS/DX30 and U-2OS/DX100 variants. These observations led us to hypothesize that snoRNA up-regulation may be associated with the first phases of acquisition of resistance, i.e., with the transition from Dox-sensitive cells, represented by U-2OS model, to moderately Dox-resistant cells, as U-2OS/DX30 and U-2OS/DX100 variants are. To validate our hypothesis, we narrowed our focus, excluding the snoRNAs with multiple chromosomal localizations and with unknown host genes. Indeed, since snoRNAs are contained in intronic sequences of multiple genes [18,20], analyzing the effects of snoRNAs localized in more than one chromosome is biased by the fact that multiple host genes are simultaneously modulated by one single snoRNA. The lack of knowledge of the host gene and the complex processing of the host mRNAs required to release snoRNAs [16,19,23] that can produce non-functioning snoRNAs and host gene mRNAs, are additional biases. In the group of snoRNAs characterized by one or two chromosomal localizations, and by a known host gene, only three, namely *SNORD3A*, *SNORA13*, and *SNORA28*, displayed a good correlation between their increase and the increase in the host gene mRNA and protein, suggesting that their processing in osteosarcoma cells was complete and produced functioning snoRNAs.

*SNORD3A* helps the processing of rRNA and its host gene *OSCP1* is a tumor suppressor gene [40]. Moreover, it encodes for a membrane protein involved in drug transport in placenta [41]. *SNORA1*3 is contained in both the 5′-UTR/promoter region and in the anti-sense sequence of *EPB41LA4*, a protein connecting cytoskeleton and plasmamembrane, and stimulating the β-catenin signaling [42]. *SNORA28* is within an intron of *EIF5*, a critical factor in the assembly of ribosomal initiation complex and GTP-driven peptide elongation [43]. To the best of our knowledge, no studies investigating the levels of *SNORA13* and *SNORA28*, nor the effects of their overexpression or silencing, have been performed in osteosarcoma and in other tumor types. *SNORD3A* and its host genes *OSCP1* were found down-regulated in uterine cervix cancer [44], although the biological and clinical meaning of this change has not been investigated. In contrast to our data, in breast cancer, *SNORD3A* overexpression chemosensitizes cells to 5-fluorouracil (5-FU), by negatively regulating miR-185-5p. The result is an increase of uridine monophosphate synthetase, the target enzyme of 5-FU [45]. We did not measure the level of endogenous miR-185-5p in osteosarcoma cells. We are aware that the pattern of miRNAs is highly variable between tumors: this factor must be considered since it may strongly impact on the biological effects of the same snoRNA in different tumor types. Interestingly the Gene Ontology analysis performed in the present study indicated Dox-related biological processes as the most strongly associated with *SNORD3A*. This is in line with previous experimental works, demonstrating that one of the mechanisms by which Dox induces cell death is the reduced biogenesis of ribosomes [27] and/or the increased ubiquitination of ribosomal proteins [28], with a consequent dramatic reduction of protein translation. On the contrary, the overexpression of proteins that restore the ribosomal efficiency, such as RRP12 [31] and RACK1 [32], is associated with Dox resistance. We thus speculated that the increase in snoRNA may be a compensatory response to prevent the ribosomal stress induced by Dox, leading to the acquisition of resistance.

In order to prove this hypothesis, we over-expressed *SNORD3A*, *SNORA1*3 and *SNORA28* in Dox-sensitive U-2OS cells. By this approach we over-expressed specific snoRNAs in a controlled way. We excluded to silence these snoRNAs in Dox-resistant variants, because the short length of snoRNAs, their redundancy and high homology could produce off-target and non-specific effects. Our results indicated that the increase in *SNORD3A*, *SNORA13*, and *SNORA28* was associated with the resistance to Dox cytotoxicity: indeed, snoRNA-over-expressing U-2OS cells have the same viability of U-2OS/DX30 and U-2OS/DX100 cells in response to Dox. This resistance was independent on the amount of the drug. The content of intracellular Dox is strictly regulated by the rate of its influx and efflux. Pgp level is the main determinant of Dox intracellular content in osteosarcoma [39]. Since *SNORD3A*, *SNORA13* and *SNORA28* did not increase the expression of Pgp in U-2OS cells, they did not change the intracellular retention of the drug that remained as high as in parental U-2OS cells and higher than in U-2OS/DX30 and U-2OS/DX100 variants. Therefore, we concluded that the lower cell death induced by Dox relies on causes independent from the intracellular drug retention.

To identify possible factors involved in this resistance, we analyzed the expression of 88 genes considered relevant for osteosarcoma pathogenesis and progression [1]: we compared the profiles of the U-2OS/DX30 and U-2OS/DX100 variants, characterized by a “patient-like” acquisition of resistance to Dox, and of snoRNA-over-expressing U-2OS cells. Three genes displayed the same modulation in all the five variants analyzed: *GADD45A* and *MYC* that were both up-regulated, and *TOP2A* that was down-regulated. The functional involvement of these three factors in Dox-resistance was proved by our reverse-genetic experiments. Indeed, the silencing of *GADD45A* and *MYC*, as well as the over-expression of *TOP2A* restored the sensitivity to Dox. This event was observed not only in *SNORD3A*-, *SNORA13*- and *SNORA28*-over-expressing U-2OS cells, but also in U-2OS/DX30 and U-2OS/DX100 variants. The results suggested that GADD45A, MYC and TOP2A may mediate the acquisition of resistance to Dox during cumulative exposure, as in the case of U-2OS/DX30 and U-2OS/DX100 variants, likely acting as down-stream targets of *SNORD3A*, *SNORA13* and *SNORA28*.

The increase in the DNA-damaging sensor GADD45A is not unexpected in U-2OS/DX30 and U-2OS/DX100 variants, which are continuously exposed to Dox, an inhibitor of topoisomerase 2 and an inducer of DNA strand breaks [46]. The acute increase of GADD45A is usually associated to apoptosis in osteosarcoma [47,48], but its prolonged increase is a compensatory response to chemotherapeutic drugs and is related to the acquisition of resistance to methotrexate [49]. MYC is a potent driver of osteosarcoma tumorigenesis [50] and its transcriptional up-regulation has been associated with resistance to Dox in solid cancers [51,52]. As for GADD45A, the increase of MYC is a marker of DNA double strand breaks and genomic instability, as well as a marker of resistance to chemotherapeutic drugs damaging DNA, because it transcriptionally activates several DNA repairing genes [53]. Moreover, MYC up-regulation increases the synthesis of rRNA and the rate of protein translation in colon cancer cells [54], thus counteracting the ribosomal stress induced by Dox. Consistently with these premises, the silencing of *GADD45A* and *MYC* chemosensitized Dox-resistant osteosarcoma cells, by disrupting adaptive responses functional to acquire Dox-resistance.

The over-expression of TOP2A is strongly associated with a good therapeutic efficacy of Dox [55]. By contrast, the decrease in *TOP2A* mRNA [56], the expression of truncated and non-functioning forms of TOP2A [57] or the enzyme inhibition [58] are all associated with resistance to Dox. Indeed, as it occurs for many chemotherapeutic drugs, mutations and/or decrease in the target make the drug less effective. In line with these findings, resistant osteosarcoma cells had a decreased expression of TOP2A, but its re-introduction restores Dox efficacy.

We propose that the continuous exposure to Dox elicits a nucleolar stress that induces a massive change in snoRNA processing, release and localization [26,30]. Surviving cells, i.e., cells acquiring resistance to Dox, up-regulate specific snoRNAs that modulate the expression of host and down-stream genes, functional to the cell survival notwithstanding the unfavorable conditions. This adaptive process leads to the progressive acquisition of resistance to Dox. The upregulation of *SNORD3A*, *SNORA13* and *SNORA28*, and the consequent modulation of *GADD45A*, *MYC*, and *TOP2A*, are prototypical examples of this mechanism (Figure 6).

Since the specific silencing of snoRNAs is at the moment a technical challenge, in a translational perspective it is more promising silencing or over-expressing the host genes. Unluckily, only in few cases the snoRNA host genes are known, limiting the number of druggable targets. At preclinical level, several siRNA- or shRNA-based tools have successfully reversed Dox resistance in osteosarcoma, e.g., by directly silencing Pgp [59,60] or by silencing transcription factors, as estrogen-related receptor α (ERRα) that up-regulates Pgp [61] or nuclear factor (erythroid-derived 2)-like 2 (NRF2) that mediates resistance to oxidative stress and up-regulates multiple ABC transporters [62]. The major advantage of these approaches is that they can achieve high specificity in targeting a gene or a pathway involved in chemoresistance. Indeed, gene therapy has been tested in phase I/phase II clinical trials in patients with solid and hematologic tumors refractory to standard treatments, obtaining satisfactory results in terms of safety and efficacy. Except four trials on Ewing’s sarcoma, no trials on osteosarcoma are active at the present (https://clinicaltrials.gov/). The main limitations of gene therapy treatments are the immunogenicity of the agents used, the presence of off-target events that produce undesired or toxic effects, the degradation of nucleic acids within the systemic circulation or the tumor microenvironment, the presence of tumor-associated barriers that limit the efficient delivery of siRNA, shRNA and gene expression vectors [63]. These limitations are not peculiar of osteosarcoma, but common to all tumors. The use of nanoparticle-based technology is a promising strategy to overcome these limitations, because it allows a controlled and actively targeting delivery of the cargo to the tumor [64]. This approach can translate the gene therapy to the clinical practice in the future.

In conclusion, we observed a surprising up-regulation of several snoRNAs in Dox-resistant variants of osteosarcoma cells. We found that *SNORD3A*, *SNORA13*, *SNORA28* are possible mediators of resistance. For the first time snoRNAs were identified as mediator of resistance to Dox in human osteosarcoma cells and as unconventional targets to induce sensitization to Dox. Validating the findings obtained in this work in patient-derived samples may help to identify novel and unexpected biomarkers predictive of response to first-line chemotherapy. Moreover, targeting specific snoRNAs or their down-stream genes may open new treatment perspectives for chemoresistant osteosarcomas that are characterized by a high risk of relapse and poor prognosis.

## 4. Materials and Methods

### 4.1. Chemicals

Fetal bovine serum (FBS) and medium for cell culture were obtained by Invitrogen Life Technologies (Carlsbad, CA, USA). Plasticware was purchased from Falcon (Becton Dickinson, Franklin Lakes, NJ, USA). The BCA kit (Sigma-Merck, St. Louis, MO, USA) was used to assess the protein content in all samples. Reagents for electrophoresis were from Bio-Rad Laboratories (Hercules, CA, USA), Dox was from Sigma-Merck. If not otherwise specified, all the other reagents were from Sigma-Merck.

### 4.2. Cells

Human Dox-sensitive osteosarcoma U-2OS and Saos-2 cells were obtained from ATCC (Manassas, VA, USA). Dox-resistant variants were established by exposing the sensitive U-2OS and Saos-2 cell lines to step-by-step increases in Dox concentrations. Cells were continuously cultured in presence of Dox during the whole selection procedure. The in vitro continuous drug exposure resulted in the establishment of variants, which were resistant to 30 ng/mL Dox (U-2OS/DX30 and Saos-2/DX30), 100 ng/mL Dox (U-2OS/DX100 and Saos-2/DX100), 580 ng/mL Dox (U-2OS/DX580 and Saos-2/DX580), corresponding to 1 µM Dox. Resistant variants were maintained in presence of the Dox concentration used for their selection [33]. Cells were grown in Iscove’s Modified Dulbecco’s Media (IMDM) (Invitrogen Life Technologies, Carlsbad, CA, USA) containing 10% *v/v* FBS, 1% *v/v* penicillin-streptomycin, 1% *v/v* L-glutamine. Cell lines were authenticated by microsatellite analysis (PowerPlex kit, Promega Corporation, Madison, WI, USA; last authentication: June 2019).

### 4.3. Gene Expression Profiling Analysis

Total RNA (300 ng) was amplified and labeled using an Illumina TotalPrep RNA Amplification Kit (Life Technologies, Carlsbad, CA, USA). A total of 750 ng of labeled cRNA probes was hybridized on the HumanHT-12 v4.0 Expression Bead Chip (Illumina, San Diego, CA, USA). Cubic spline-normalized probe intensity data, together with detection P-values, were obtained using the GenomeStudio software v2011.01 (Illumina). Probes were selected only if at least one experimental point showed a detection *p*-value < 0.05. For each gene, we retained the associated probe with the largest mean expression value across all samples. Differential expression analysis was performed using the eBayes function of the LIMMA R package [65]. A gene was defined differentially expressed if associated with a *p*-value < 0.01. The association of snoRNA and target genes was obtained correlating their expression and selecting those with an absolute Pearson r value >0.9. The gene set enrichment analysis was performed using the Enrichr tools [66].

### 4.4. SnoRNA Overexpression

Expression vectors for *SNORD3A*, *SNORA13* and *SNORA28* were generated as follows. SnoRNA inserts were amplified by PCR from 100 ng of genomic DNA using PfuTurbo DNA Polymerase AD (Agilent Technologies, Santa Clara, CA, USA). The PCR product was cut with BamhI and XhoI restriction enzymes and cloned into the BamhI-XhoI sites of the pcDNA3 vector. Oligonucleotides are specified in Table 1 (cut sites are underlined). All vectors were sequenced before use.

Two × 10^5^ cells were grown until 70–80% confluence, then transfected with 1 µg of specific pcDNA3 plasmids using jetPRIME transfection reagent (Polyplus, New York, NY, USA), as per manufacturer’s instructions. SnoRNA overexpressing cells were selected with 800 mg/L G418 for 4 days, then maintained in culture medium containing 200 mg/L G418.

### 4.5. Cell Silencing/Overexpression

Briefly, two × 10^5^ cells were grown until 70–80% confluence. In silencing experiments, cells were transfected with 0.5 µg of Control non-targeting siRNA plasmid (sc-37007); GADD 45α siRNA plasmid (sc-35440); c-Myc siRNA plasmid (sc-29226) (all from Santa Cruz Biotechnology Inc., Santa Cruz, CA, USA). In over-expression experiments, cells were transfected with 1 µg of Control (empty) CRISPR Activation Plasmid or Topo IIα CRISPR Activation Plasmid (sc-400869-ACT Santa Cruz Biotechnology Inc.), according to the manufacturer’s instructions. The efficacy of silencing or overexpression was controlled by RT-PCR (24 h after the transfection) or by immunoblot (48 h after the transfection). For the experiments. cells were used 48 h after the transfection.

### 4.6. RT-PCR and PCR Array

Total RNA was extracted and reverse-transcribed using iScript^TM^ cDNA Synthesis Kit (Bio-Rad Laboratories). The RT-PCR was performed with the IQ SYBR Green Supermix (Bio-Rad Laboratories). The primer sequences are listed in Table 2. The relative quantitation was performed by comparing each PCR product with the housekeeping PCR product *S14*, using the Bio-Rad Software Gene Expression Quantitation (Bio-Rad Laboratories). The PCR arrays were performed on 1 μg cDNA, using the Osteosarcoma PCR Array (Bio-Rad Laboratories), as per manufacturer’s instructions.

### 4.7. Doxorubicin Intracellular Accumulation

The amount of intracellular Dox was detected fluorimetrically [39], using a Synergy HT 96-well micro-plate reader (Bio-Tek Instruments, Winooski, VT, USA). Excitation and emission wavelengths were 475 and 553 nm, respectively. A blank was prepared in the absence of cells in each set of experiments and its fluorescence was subtracted from that measured in each sample. Fluorescence was converted in nmoles doxorubicin/mg cell proteins using a calibration curve.

### 4.8. Immunoblot

Cell lysates were prepared by rinsing cells with 0.5 mL of lysis buffer (125 mM Tris-HCl, 750 mM NaCl, 1% *v/v* NP40, 10% *v/v* glycerol, 50 mM MgCl_2_, 5 mM EDTA, 25 mM NaF, 1 mM Na_3_VO_4_, 10 mg/mL leupeptin, 10 mg/mL pepstatin, 10 mg/mL aprotinin, 1 mM phenylmethylsulphonyl fluoride, pH 7.5). Samples were sonicated and centrifuged at 13,000× g at 4 °C for 10 min. The immunoblot analyses were performed on 50 μg of proteins, using the following antibodies: anti-OSCP1 (ab244416, Abcam, Cambridge, UK; dilution 1/2000), anti-EPB41L4A (ab67551, Abcam; dilution 1/500), anti-EIF5 (ab228874, Abcam; dilution 1/1000), anti-ABCB1 (C219, Novus Biologicals, Littleton, CO; dilution 1/250), anti-GADD45A (ab180768, Abcam; dilution 1/500), anti-MYC (10828-1-AP, Proteintech Group Inc., Rosemont, IL, USA; dilution 1/1000), anti-TOP2A (20233-1-AP, Proteintech Group Inc.; dilution 1/1000), followed by a peroxidase-conjugated secondary antibody. Anti-β-tubulin antibody (sc-5274, Santa Cruz Biotechnology Inc.; dilution 1/1000) was used as control of equal protein loading. The proteins were detected by enhanced chemiluminescence (Bio-Rad Laboratories). The relative quantitation of immunoblot was performed with the ImageJ software (https://imagej.nih.gov/ij/; v1.52t). The band density of untreated cells was considered as 1 arbitrary unit.

### 4.9. LDH Release

The release of LDH, considered an index of cell damage and necrosis, was measured as reported in [67]. The results were expressed as percentage of extracellular LDH versus total (intracellular plus extracellular) LDH.

### 4.10. Cell Viability

One × 10^5^ cells were stained with 5% *w/v* crystal violet solution in 66% *v/v* methanol, washed and photographed. Quantitation of crystal violet staining was performed after solubilizing the dye and reading the absorbance of each well at 540 nm (HT Synergy 96-well micro-plate reader). The mean absorbance of untreated cells was considered 100% and the absorbance units were expressed as percentage of viable cells vs. untreated cells.

### 4.11. Statistical Analysis

Results were analyzed by a one-way analysis of variance (ANOVA) and Tukey’s test, using GraphPad Prism software (v6.01, San Diego, CA, USA). *p* < 0.05 was considered significant. All data were expressed as means ± SD.

## Figures and Tables

**Figure 1 ijms-21-04500-f001:**
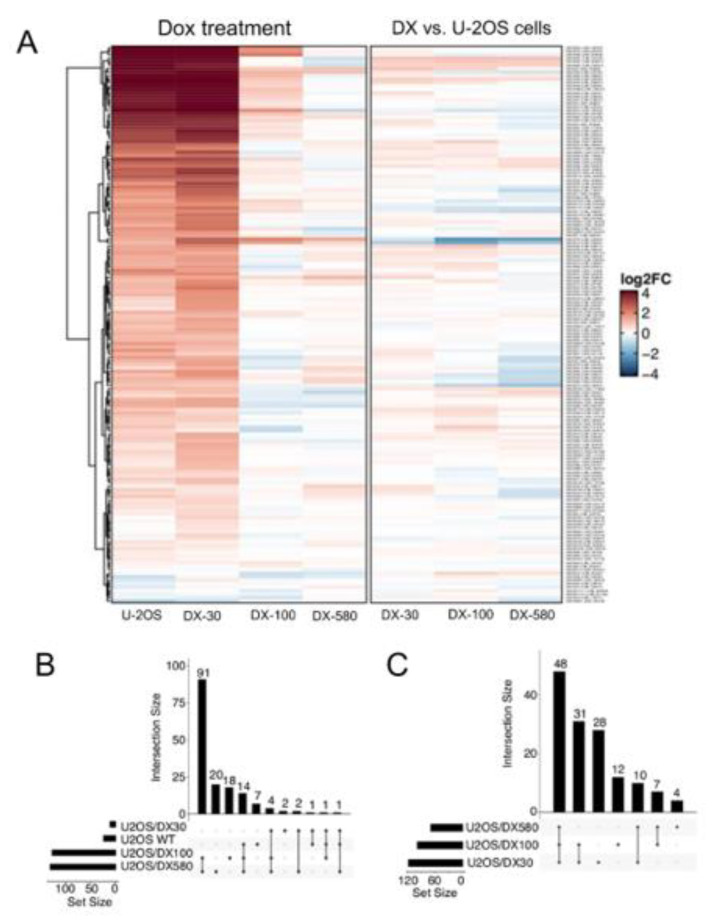
snoRNA family is up-regulated upon doxorubicin treatment and in doxorubicin-resistant osteosarcoma cells. (**A**) Heatmap of snoRNA expression changes upon 5 μM Dox treatment for 24 h, in Dox-sensitive U-2OS cells and Dox-resistant variants (U-2OS/DX30, U-2OS/DX100 and U-2OS/DX580) (left) or computed between U-2OS cells and resistant variants (right). (**B**) UpSet plot reporting the overlap between snoRNA significantly and differentially expressed upon Dox treatment in sensitive and resistant cells (**C**) UpSet plot reporting the overlap between snoRNA upregulated in Dox-resistant variants compared to Dox-sensitive cells. The results are the means of two independent experiments.

**Figure 2 ijms-21-04500-f002:**
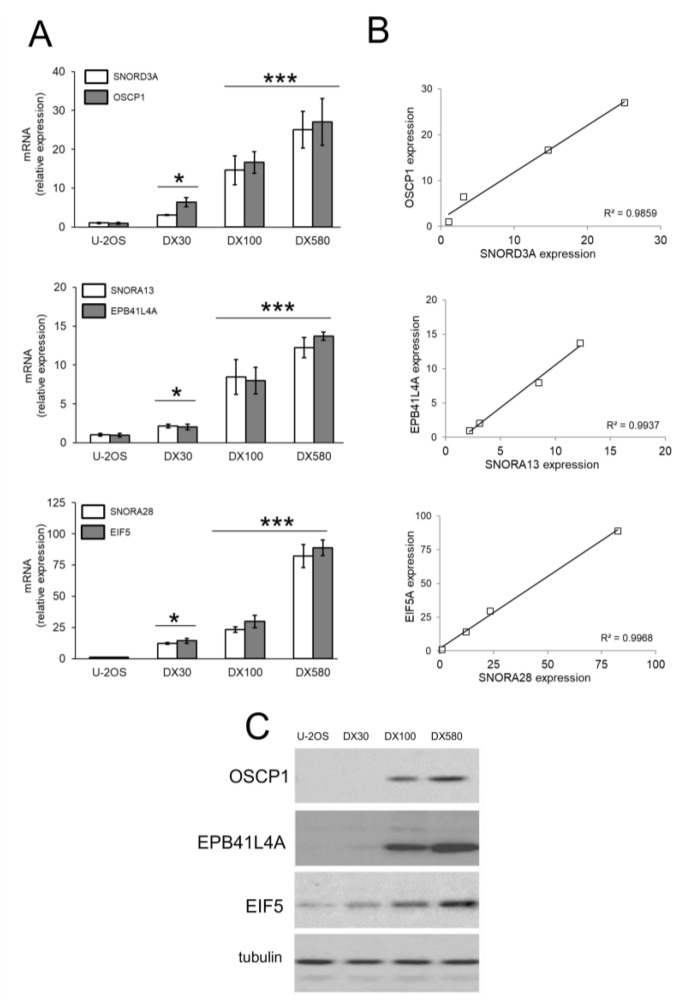
*SNORD3A*, *SNORA13* and *SNORA28* up-regulation in doxorubicin-resistant osteosarcoma cells. (**A**) mRNA levels of *SNORD3A*, *SNORA13* and *SNORA28* and their host genes (*OSCP1*, *EPB41L4A* and *EIF5*) were evaluated by RT-PCR, in triplicates. Data are means ± SD (*n* = 3). * *p* < 0.05, *** *p* < 0.001: DX-variants vs. U-2OS cells. (**B**) Linear correlation between the relative expression of snoRNAs and the relative expression of the host genes, according to the RT-PCR results of Figure 2A. (**C**) Immunoblot of the indicated proteins. The figure is representative of one out of three experiments. Tubulin was used as control of equal protein loading.

**Figure 3 ijms-21-04500-f003:**
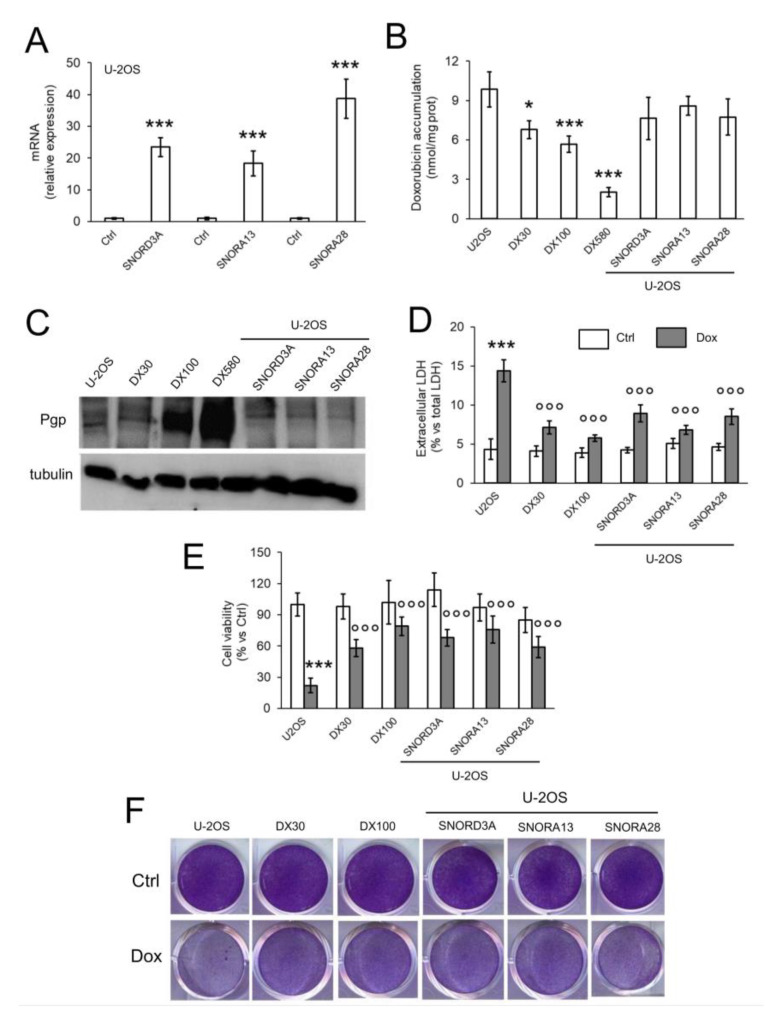
*SNORD3A*, *SNORA13* and *SNORA28* induce resistance to doxorubicin in sensitive osteosarcoma cells. U-2OS cells were transfected with empty vector (Ctrl) or expression vectors for *SNORD3A*, *SNORA13* and *SNORA28*. When indicated, cells were treated with 5 μM Dox for 3 h (panel **B**), 24 h (panel **D**), 72 h (panels **E**,**F**). U-2OS/DX30, U-2OS/DX100 and U-2OS/DX580 cells were used as control of Dox-resistant cells. (**A**) mRNA levels of *SNORD3A*, *SNORA13* and *SNORA28*, evaluated by RT-PCR, in triplicates. Data are means + SD (*n* = 3). *** *p* < 0.001: snoRNA-over-expressing U-2OS cells vs. Ctrl cells. (**B**) Intracellular doxorubicin accumulation, measured fluorometrically, in duplicates. Data are means ± SD (*n* = 3). * *p* < 0.05, *** *p* < 0.001: snoRNA-over-expressing U-2OS cells vs. Ctrl cells. (**C**) Pgp immunoblot. The figure is representative of one out of three experiments. Tubulin was used as control of equal protein loading. (**D**) Release of LDH, an index of cell necrosis, measured spectrophotometrically, in duplicates. Data are means + SD (*n* = 3). *** *p* < 0.001: Dox-treated U-2OS cells vs. Ctrl cells; °°° *p* < 0.001: U-2OS/DX30, U-2OS/DX100 or snoRNA-over-expressing U-2OS cells vs. Ctrl U-2OS cells. (**E**) Cell viability spectrophotometric quantification after crystal violet staining, used as assay of cell viability. Data are means + SD (*n* = 4). *** *p* < 0.001: Dox-treated U-2OS cells vs. Ctrl cells; °°° *p* < 0.001: U-2OS/DX30, U-2OS/DX100 or snoRNA-over-expressing U-2OS cells vs. Ctrl U-2OS cells. (**F**) Representative photographs of crystal violet staining. The photograph is representative of one out of four experiments, in quadruplicates.

**Figure 4 ijms-21-04500-f004:**
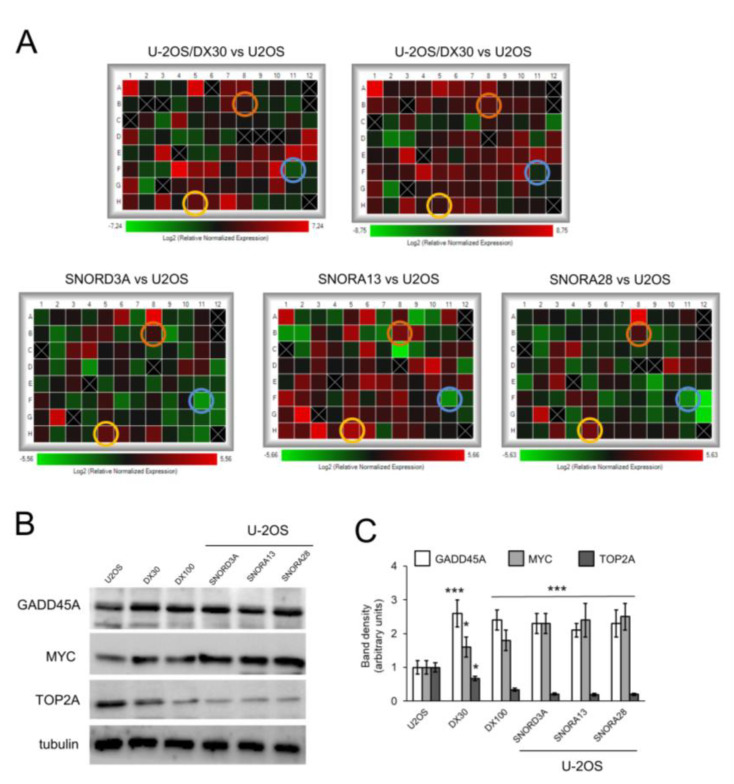
*SNORD3A*, *SNORA13* and *SNORA28* up-regulate *GADD45A* and *MYC*, downregulate *TOP2A*. U-2OS cells were transfected with empty vector (U-2OS) or expression vectors for *SNORD3A*, *SNORA13* and *SNORA28*. U-2OS/DX30, U-2OS/DX100 and U-2OS/DX580 cells were used as control of Dox-resistant cells. (**A**) Heatmaps of changes in the expression of genes relevant for osteosarcoma pathogenesis and progression, measured by PCR arrays (*n* = 4). Yellow circles: *GADD45A* hits; orange circles: *MYC* hits; blue circles: *TOP2A* hits. (**B**) Immunoblot of the indicated proteins. The figure is representative of one out of three experiments. Tubulin was used as control of equal protein loading. (**C**) Immunoblot quantitation. The quantitation of the band density was performed using the ImageJ software. Data are means + SD (*n* = 3). * *p* < 0.05, *** *p* < 0.001: U-2OS/DX30, U-2OS/DX100 or snoRNA-over-expressing U-2OS cells vs. parental U-2OS cells.

**Figure 5 ijms-21-04500-f005:**
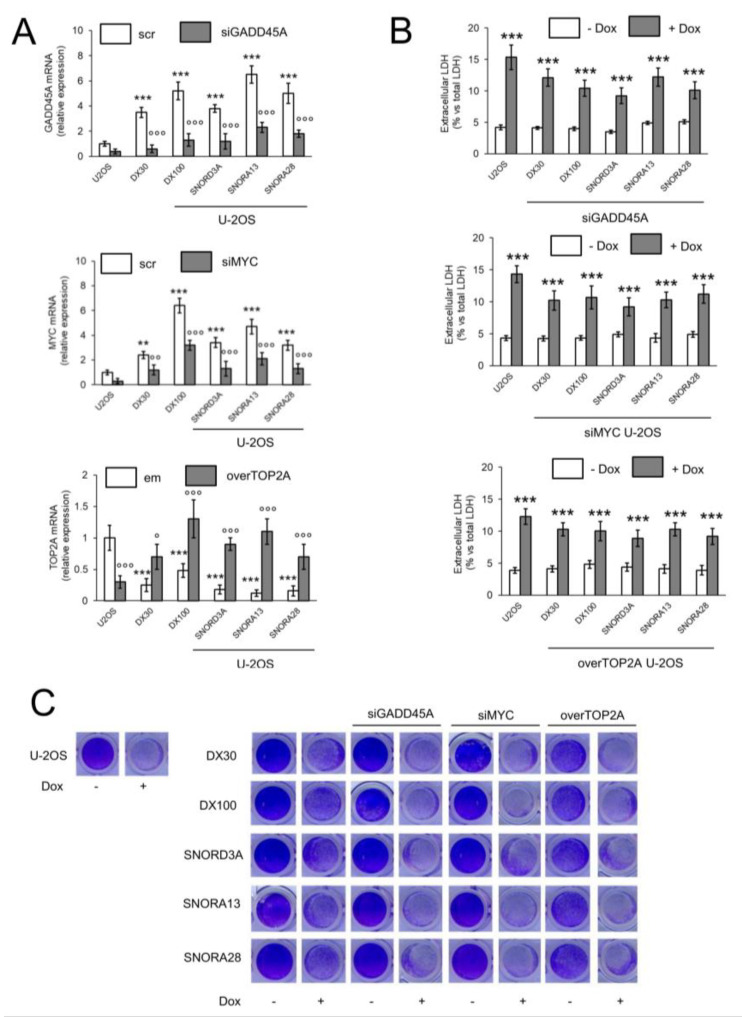
*GADD45A*, *MYC* and *TOP2A* are responsible for the doxorubicin resistance induced by *SNORD3A*, *SNORA13* and *SNORA28*. U-2OS cells were transfected with an expression vector for *SNORD3A*, *SNORA13* or *SNORA28*. When indicated, cells were transiently transfected with a non-targeting scrambled siRNA pool (scr), with a *GADD45A*- or *MYC*-targeting siRNA (siGADD45A/siMYC) pool, with an empty vector (em) or with an expression vector for *TOP2A* (overTOP2A). U-2OS/DX30 and U-2OS/DX100 cells were included as Dox-resistant cells. Cells were treated with 5 μM Dox for 24 h (panel **B**) or 72 h (panel **C**). (**A**) mRNA levels of *GADD45A*, *MYC* and *TOP2*A, evaluated by RT-PCR, in triplicates. Data are means ± SD (*n* = 3). ** *p* < 0.01, *** *p* < 0.001: U-2OS/DX30, U-2OS/DX100 or snoRNA-over-expressing U-2OS cells vs. U-2OS cells; °° *p* < 0.01, °°° *p* < 0.001: siGADD45A/siMYC vs. scr cells, overTOP2A vs. em cells. (**B**) Release of LDH, an index of cell necrosis, measured spectrophotometrically, in duplicates. Data are means ± SD (*n* = 3). *** *p* < 0.001: Dox-treated U-2OS cells vs. untreated (-Dox) cells. (**C**) Representative photographs of crystal violet staining, used as assay of cell viability. The photograph is representative of one out of four experiments, in quadruplicates.

**Figure 6 ijms-21-04500-f006:**
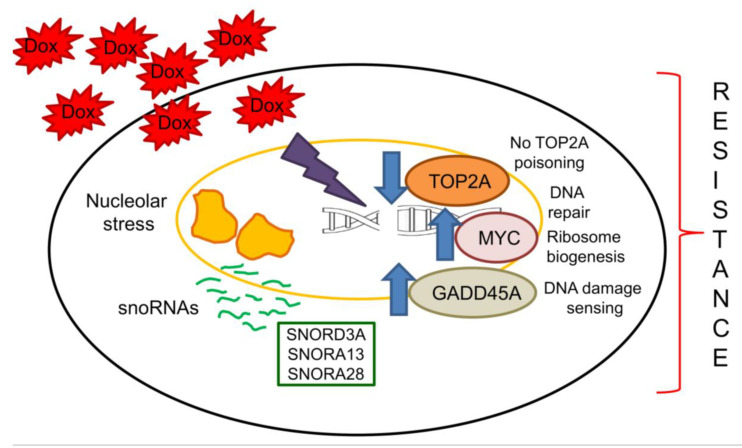
Linkage between snoRNAs and acquisition of resistance to doxorubicin. The exposure to Dox induces a continuous nucleolar stress that up-regulates several snoRNAs, including *SNORD3A*, *SNORA13* and *SNORA28*. Their increase mediates resistance to Dox by down-regulating *TOP2A*, the main target of Dox, and by up-regulating *GADD45A*, a DNA damaging sensor that promotes the adaptation to chronic stresses, and *MYC*, a strong oncogenic factor inducing DNA repair genes and ribosomal protein synthesis. Arrow: decrease or increase, respectively.

**Table 1 ijms-21-04500-t001:** Oligonucleotide for snoRNA expression vectors.

Gene	Primers
*SNORD3A_BAMHI_F*	GCGGATCCAAGACTATACTTTCAGGGATCATTTC
*SNORD3A_XHOI_R*	CGCTCGAGACCACTCAGACCGCGTT
*SNORA13_BAMHI_F*	GCGGATCCAGCCTTTGTGTTGCC
*SNORA13_XHOI_R*	CGCTCGAGAACTGTTACTTATGCAGCTCC
*SNORA28_BAMHI_F*	GCGGATCCAAGCAACACTCTGTG
*SNORA28_XHOI_R*	CGCTCGAGACTGTTTAAGTCTATATAACGGC

Underlined sites: cutting sites.

**Table 2 ijms-21-04500-t002:** Primer list.

Gene	Primers
*SNORD3A_F*	TAGAGCACCGAAAACCACGA
*SNORD3A_R*	CCTCTCACTCCCCAATACGG
*OSCP1_F*	ATCAACATACAAGCCACCCAG
*OSCP1_R*	ATCATGGCGAGCAAATCGTC
*SNORA13_F*	TTTGTGTTGCCCATTCACTTTG
*SNORA13_R*	ACTTATGCAGCTCCTACACCAA
*EPB41L4A_F*	AAGCAGCGTGCCTGGTTAC
*EPB41L4A_R*	ATGCTCCCAGATGGTATTCAGC
*SNORA28_F*	GTCTGACACAATTTGAGCTTGCT
*SNORA28_R*	ATAACGGCTTGTCTCATGGGA
*EIF5_F*	CACCTGAGAATAGTGACAGTGGT
*EIF5_R*	TCATTTGGTGGTGGTGGTGG
*GADD45A_F*	TGCTCAGCAAAGCCCTGAGT
*GADD45A_R*	GCAGGCACAACACCACGTTA
*MYC_F*	ACCACCAGCAGCGACTCTGA
*MYC_R*	TCCAGCAGAAGGTGATCCAGACT
*TOP2A_F*	CTAGTTAATGCTGCGGACAACA
*TOP2A_R*	CATTTCGACCACCTGTCACTT
*S14_F*	CGAGGCTGATGACCTGTTCT
*S14_R*	GCCCTCTCCCACTCTCTCTT

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
