# Peer review of "Small Nucleolar RNAs Determine Resistance to Doxorubicin in Human Osteosarcoma"

_ijms, 2020, doi:10.3390/ijms21124500_

Round 1

Reviewer 1 Report

The aim of this manuscript was to study and identify new markers associated with doxorubicin resistance. Particularly, the authors investigated if and how are small nuclear RNAs (snoRNAs) linked to resistance in osteosarcoma sensitive and resistant cells. The authors reported up-regulated snoRNAs in osteosarcoma cells with resistance and they found that those snoRNAs induce resistance to Dox by modulating the expression of genes involved in DNA repair system and proliferation. The theme of the role of snoRNAs in therapy response is very actual and they perform new potential targets in the therapy of chemoresistant osteosarcomas. This manuscript is capable of being published after the minor revision process;

  1. Introduction part; the last paragraph, authors should mention the aims of the study instead of conclusions of this manuscript. This paragraph summarizing findings of the study should be included at the end of the manuscript.
  2. Results, paragraph 2.1.; There is not mentioned if the authors performed biological doubles of snoRNA expression arrays.
  3. Discussion part; Authors should mention a few facts describing modifications of particular snoRNAs mentioned in the manuscript in different tumor cells if these manipulations are known and investigated.

Author Response

The aim of this manuscript was to study and identify new markers associated with doxorubicin resistance. Particularly, the authors investigated if and how are small nuclear RNAs (snoRNAs) linked to resistance in osteosarcoma sensitive and resistant cells. The authors reported up-regulated snoRNAs in osteosarcoma cells with resistance and they found that those snoRNAs induce resistance to Dox by modulating the expression of genes involved in DNA repair system and proliferation. The theme of the role of snoRNAs in therapy response is very actual and they perform new potential targets in the therapy of chemoresistant osteosarcomas. This manuscript is capable of being published after the minor revision process;

1. Introduction part; the last paragraph, authors should mention the aims of the study instead of conclusions of this manuscript. This paragraph summarizing findings of the study should be included at the end of the manuscript.

We modified the Introduction highlighting that the aim of the study was identifying possible cause-effect relations between the expression of specific snoRNAs and the resistance to Dox (line 114). As suggested, we moved the last sentence of the Introduction to the end of the manuscript (line 383).

2. Results, paragraph 2.1.; There is not mentioned if the authors performed biological doubles of snoRNA expression arrays.

We apologize for not having reported that the results are the mean of two independent experiments. We added this information in the legend of Figure 1.

3. Discussion part; Authors should mention a few facts describing modifications of particular snoRNAs mentioned in the manuscript in different tumor cells if these manipulations are known and investigated.

To the best of our knowledge, no studies investigating the levels of SNORA13 and SNORA28, nor the effects of their overexpression or silencing, have been performed in osteosarcoma and in other tumor types. SNORD3A and its host genes OSCP1 were found down-regulated in uterine cervix cancer [Roychowdhury, 2020], although the biological and clinical meaning of this change has not been investigated. In contrast to our data, in breast cancer, SNORD3A overexpression chemosensitizes cells to 5-fluorouracil (5-FU), by negatively regulating miR-185-5p. The result is an increase of uridine monophosphate synthetase, the target of 5-FU [Luo, 2020]. We did not measured the level of endogenous miR-185-5p in osteosarcoma cells. The pattern of miRNAs is highly variable between tumors; this factor must be considered since it may strongly impact on the biological effects of the same snoRNA in different tumor types.

We mentioned this point in the Discussion (line 287). We added two new references.

Reviewer 2 Report

IJMS 837014_v1 Kopecka

The manuscript submitted by Dr. Janna Kopecka reports up-regulation of the SNORD3A, SNORA13, and SNOR28 snoRNAs in osteosarcoma cells following selection for moderate resistance to doxorubicin (Dox). Overexpression of the snoRNAs in sensitive cells increased Dox resistance strongly suggesting a causative relationship between the snoRNAs and resistance. Furthermore, GADD45A and MYC were up-regulated, and TOP2A was down-regulated in the resistant cells. This indicated that the three snoRNAs are useful markers for Dox resistance in osteosarcoma cells, which is significant since Dox often is used as a first-line drug in chemotherapy. The authors discuss various possible mechanisms, but the discussion is, in my view, incomplete.

  1. The authors mention that the three snoRNAs are known to assist the modification of ribosome formation and function. However, they do not consider this as a possible contributor to the DOX resistance. I suggest they give this some thought during the revision of the manuscript. MYC is known to stimulate rRNA transcription, and as mentioned by the authors, the snoRNAs are known as ribosome assembly/maturation factors. Furthermore, it has been reported that Doxorubicin-induced DNA damage causes extensive ubiquitination of ribosomal proteins (DOI: 1074/mcp.RA118.000652). Taken together with the fact that tumor cells typically depend on upregulation of ribosome formation, it may be reasonable to think that the increased resistance to Dox is facilitated by an increased ribosome concentration.
  2. Another interesting connection is that the Dox chemical structure has strong similarities to tetracycline which inhibits bacterial ribosomes. As far as I know, Dox has not been identified as a ribosomal antibiotic, but I am not aware of any direct analysis either.
  3. The manuscript should be reviewed for English syntax and typos, for example
    • Line 43: In the last years> In recent years
    • Line 73: C0 and D0> C’ and D’. This may be a result of the conversion to pdf format

Author Response

The manuscript submitted by Dr. Janna Kopecka reports up-regulation of the SNORD3A, SNORA13, and SNOR28 snoRNAs in osteosarcoma cells following selection for moderate resistance to doxorubicin (Dox). Overexpression of the snoRNAs in sensitive cells increased Dox resistance strongly suggesting a causative relationship between the snoRNAs and resistance. Furthermore, GADD45A and MYC were up-regulated, and TOP2A was down-regulated in the resistant cells. This indicated that the three snoRNAs are useful markers for Dox resistance in osteosarcoma cells, which is significant since Dox often is used as a first-line drug in chemotherapy. The authors discuss various possible mechanisms, but the discussion is, in my view, incomplete.

  1. The authors mention that the three snoRNAs are known to assist the modification of ribosome formation and function. However, they do not consider this as a possible contributor to the DOX resistance. I suggest they give this some thought during the revision of the manuscript. MYC is known to stimulate rRNA transcription, and as mentioned by the authors, the snoRNAs are known as ribosome assembly/maturation factors. Furthermore, it has been reported that Doxorubicin-induced DNA damage causes extensive ubiquitination of ribosomal proteins (DOI: 1074/mcp.RA118.000652). Taken together with the fact that tumor cells typically depend on upregulation of ribosome formation, it may be reasonable to think that the increased resistance to Dox is facilitated by an increased ribosome concentration.

We thanks the Reviewer for this interesting comment.

Following the suggestion, we implemented the manuscript reporting the experimental evidences that Dox is a potent strong inducer of nucleolar stress. Indeed the DNA damage elicited by the drug disaggregates the nucleolar structure, impairs the ribosome biogenesis occurring in nucleolus, promotes the translocation of the cytoprotective protein nucleophosmin from nucleolus to cytoplasm [Lu, 2018]. Dox exposure also activates a broad transcriptional program that decreases aminoacid biosynthesis, tRNA aminoacylation and general translation [Nikerel, 2018]. In human osteosarcoma U-2OS cells, Dox-induced DNA damages increase the ubiquitination of ribosomal proteins of both 40S and 60S subunits [Halim, 2018]. The resulting decrease in ribosome biogenesis and/or increases in ribosomal protein degradations dramatically reduce protein translation and trigger apoptotic cell death. Dox also induces the cytoplasmatic export of C/D snoRNAs contained in the introns of the ribosomal protein L13a (Rpl13a), a physiological contributor of cell death in response to oxidative stress [Holley, 2015]. By contrast, resistance to Dox has been associated to the increased expression of specific ribosome components. In U-2OS cells, Dox-induced DNA damage and cell cycle arrest decrease the expression of the nucleolar protein ribosomal RNA processing 12 homolog (RRP12), which is involved in the export and maturation of 40S and 60S ribosome components. By contrast, RRP12 overexpression determines Dox resistance [Choi et al, 2016]. Similarly, the aberrant expression of the receptor for activated C-kinase 1 (RACK1) and its ribosomal localization induce resistance to Dox, by phosphorylating the eukaryotic initiation factor 4E (eIF4E) that triggers the preferential translation of pro-survival factors [Ruan, 2012]. (Introduction, line 92).

We further commented these findings and the results obtained in the present work in the Discussion (line 298). Since the Gene Ontology analysis (Supplementary Tables S2-S3) showed Dox-related biological processes as the most strongly associated with SNORD3A, that helps the processing of rRNA, we speculated that the increase in this snoRNA may be a compensatory response to prevent the ribosomal stress induced by Dox, leading to the acquisition of resistance.

We also thanks the Reviewer for the suggestion of linking MYC to ribosome functions. In the Discussion (line 339), we included the findings reporting that MYC up-regulation increases the synthesis of rRNA and the rate of protein translation in colon cancer cells [Morcelle, 2019]. This mechanism can counteract the ribosomal stress induced by Dox. Moreover, it may explain why Myc is up-regulated in osteosarcoma cells with acquired resistance to Dox, while Myc-silencing restores Dox sensitivity.

We modified the Abstract (line 44), the Figure 6 and its legend accordingly.

We added seven new references.

  1. Another interesting connection is that the Dox chemical structure has strong similarities to tetracycline which inhibits bacterial ribosomes. As far as I know, Dox has not been identified as a ribosomal antibiotic, but I am not aware of any direct analysis either.

We thanks the Reviewer for pointing out the parallelism between tetracycline and Dox. To the best of our knowledge, however, the mechanism of action of the two drugs are different.

Tetracycline inhibits tRNA attachment to the 30S subunit of bacterial ribosomes, inhibiting the elongation of the nascent peptide. Dox inhibits ribosomal functions in an indirect way. Besides reducing ribosome biogenesis [Lu, 2018] and tRNA aminoacylation [Nikerel, 2018], and increasing ribosomal protein ubiquitination [Halim, 2018], Dox inhibits mammalian target of rapamycin (mTOR) activity: as a consequence, the phosphorylation of p70 S6 kinase (p70 S6K) and eukaryotic initiation factor 4E-binding protein 1 (eIF4E-BP1) are reduced [Gaur, 2011]. These events impair the p70 S6K-mediated phosphorylation of ribosomal proteins and sequester the cap-binding protein eIF4E, respectively, decreasing the initiation of protein translation.

We included this paragraph in the Introduction (line 98) and we added one new reference.

  1. The manuscript should be reviewed for English syntax and typos, for example
    • Line 43: In the last years> In recent years
    • Line 73: C0 and D0> C’ and D’. This may be a result of the conversion to pdf format

We reviewed English syntax and corrected the typos throughout the manuscript.

Reviewer 3 Report

Doxorubicin (Dox) is one of the most important first-line drugs used in osteosarcoma therapy. The results indicated a global up-regulation of small nucleolar RNAs (snoRNAs) in human Dox-resistant osteosarcoma cells. In addition, snoRNAs up-regulated in osteosarcoma cells with different degrees of resistance to Dox. Furthermore, SNORD3A, SNORA13 and SNOR28 reduced Dox-cytotoxicity when over-expressed in Dox-sensitive cells. In these cells, GADD45A and MYC were up-regulated, TOP2A was down-regulated. Therefore, targeting snoRNAs or down-stream genes may open new treatment perspectives in chemoresistant osteosarcomas.

Major points:

  1. The detail method of established Dox-resistant cells should be described in Method.
  2. Fig. 2. The protein expression should be confirm the results from mRNA
  3. Fig. 3F. The images are not clear. It's very difficult to read.
  4. The clinic application should be added in Discussion
  5. The limitation should be discussed.
  6. PDGF and PDGFR are important targeting of human sarcoma. Are there involved in Dox effect in osteosarcoma?

Author Response

Doxorubicin (Dox) is one of the most important first-line drugs used in osteosarcoma therapy. The results indicated a global up-regulation of small nucleolar RNAs (snoRNAs) in human Dox-resistant osteosarcoma cells. In addition, snoRNAs up-regulated in osteosarcoma cells with different degrees of resistance to Dox. Furthermore, SNORD3A, SNORA13 and SNOR28 reduced Dox-cytotoxicity when over-expressed in Dox-sensitive cells. In these cells, GADD45A and MYC were up-regulated, TOP2A was down-regulated. Therefore, targeting snoRNAs or down-stream genes may open new treatment perspectives in chemoresistant osteosarcomas.

Major points:

1. The detail method of established Dox-resistant cells should be described in Method.

We detailed the method of selection of Dox-resistant variants (Materials and Methods, line 398).

2. Fig. 2. The protein expression should be confirm the results from mRNA

Following the suggestions of the Reviewer, we measured the levels of OSCP1, EPB41L4A and EIF5 by immunoblotting. As shown in the new Figure 2C, the changes in protein expression followed the changes in mRNA levels. We modified Results (line 151), Discussion (line 280), Materials and Methods (line 460) and the legend of Figure 2 accordingly.

3. Fig. 3F. The images are not clear. It's very difficult to read.

We apologize for the inconvenience. We provided an image with higher resolution in the revised manuscript.

4. The clinic application should be added in Discussion

Several siRNA- or shRNA-based tools have successfully reversed Dox resistance in osteosarcoma preclinical models, e.g. by silencing Pgp [Susa, 2010; Perez, 2011] or by silencing transcription factors, as estrogen-related receptor α (ERRα) that up-regulates Pgp [Chen, 2018] or nuclear factor (erythroid-derived 2)-like 2 (NRF2) that mediates resistance to oxidative stress and up-regulates multiple ABC transporters [Li, 2018]. The major advantage of these tools is that they can achieve high specificity in targeting a gene or a pathway involved in chemoresistance. Indeed, gene therapy has been tested in phase I/phase II clinical trials in patients with solid and hematologic tumors refractory to standard treatments, obtaining satisfactory results in terms of safety and efficacy. Except four trials on Ewing’s sarcoma, no trials on osteosarcoma are active at the present (https://clinicaltrials.gov/). This can be due to the low frequency of osteosarcoma and to the general limitations of gene therapy-based treatments (see reply to point 5), not to a peculiar phenotype of osteosarcoma.

We added a paragraph in the Discussion (line 363) and four new references.

5. The limitation should be discussed.

We agree that two types of limitations exist.

First, the specific silencing of snoRNA is at the moment a technical challenge; in a translational perspective it is more promising to silence or over-express their target genes, but they are not always known. Second, the main limitations of gene therapy treatments, common to all solid and hematologic cancers, are represented by the immunogenicity of the agents used, by the presence of off-target events that produce undesired or toxic effects, by the degradation of nucleic acids within the systemic circulation or tumor microenvironment, by the presence of tumor-associated barriers that limit the efficient delivery of siRNA, shRNA and gene expression vectors [Senapati, 2019]. The use of nanoparticle-based technology is a promising strategy to overcome these limitations, because it allows a controlled and actively targeted delivery of their cargo to the tumor [Jin, 2020]. This approach can translate the gene therapy to the clinical practice in the future.

We included this paragraph in the Discussion (line 374). We added two new references.

6. PDGF and PDGFR are important targeting of human sarcoma. Are there involved in Dox effect in osteosarcoma?

We agree that the PDGF/PDGFR signalling plays a critical role in osteosarcoma development and metastasis [Xu, J. et al, Clin Sarcoma Res. 2018, 8, 15. doi:10.1186/s13569-018-0102-1], but to the best of our knowledge it has not been correlated with Dox effects. Only one study reported that the PDGFR axis inhibitor imatinib was ineffectively if used alone, but if combined with Dox it obtained a synergistic anti-tumor effect in preclinical models of osteosarcoma. However, the anti-tumor efficacy reached by the combination treatment was not significantly different from the efficacy reached by Dox alone [Yamaguchi, S.I. et al, Cancer Sci 2015, 106, 875–882. doi:10.1111/cas.12686]. This result indicates that Dox may enhance the therapeutic benefit of imatinib, but it is unlikely that PDGF/PDGFR signalling affects Dox sensitivity or resistance.

Round 2

Reviewer 3 Report

Accept to publish